# Fabrication of Pressure Sensor Using Electrospinning Method for Robotic Tactile Sensing Application

**DOI:** 10.3390/nano11051320

**Published:** 2021-05-17

**Authors:** Tamil Selvan Ramadoss, Yuya Ishii, Amutha Chinnappan, Marcelo H. Ang, Seeram Ramakrishna

**Affiliations:** 1Department of Mechanical Engineering, Faculty of Engineering, National University of Singapore, 9 Engineering Drive 1, Singapore 117575, Singapore; mpecam@nus.edu.sg (A.C.); mpeangh@nus.edu.sg (M.H.A.); seeram@nus.edu.sg (S.R.); 2Faculty of Fiber Science and Engineering, Kyoto Institute of Technology, Kyoto 606-8585, Japan; yishii@kit.ac.jp

**Keywords:** electrospinning, non-piezoelectric polymers, tactile sensors, robotic gripper

## Abstract

Tactile sensors are widely used by the robotics industries over decades to measure force or pressure produced by external stimuli. Piezoelectric-based pressure sensors have intensively been investigated as promising candidates for tactile sensing applications. In contrast, piezoelectric-based pressure sensors are expensive due to their high cost of manufacturing and expensive base materials. Recently, an effect similar to the piezoelectric effect has been identified in non-piezoelectric polymers such as poly(d,l-lactic acid (PDLLA), poly(methyl methacrylate) (PMMA) and polystyrene. Hence investigations were conducted on alternative materials to find their suitability. In this article, we used inexpensive atactic polystyrene (aPS) as the base polymer and fabricated functional fibers using an electrospinning method. Fiber morphologies were studied using a field-emission scanning electron microscope and proposed a unique pressure sensor fabrication method. A fabricated pressure sensor was subjected to different pressures and corresponding electrical and mechanical characteristics were analyzed. An open circuit voltage of 3.1 V was generated at 19.9 kPa applied pressure, followed by an integral output charge (Δ*Q*), which was measured to calculate the average apparent piezoelectric constant *d*_app_ and was found to be 12.9 ± 1.8 pC N^−1^. A fabricated pressure sensor was attached to a commercially available robotic arm to mimic the tactile sensing.

## 1. Introduction

Tactile sensing technology has undoubtedly attracted more attention in the field of robotics. These sensors have been looked upon as the best choice for improving the precision and contact with a robot’s environment [1,2,3]. Tactile sensing devices, as shown in big 1a, which emulate the properties of human skin, in other words electronic skin structures, have been widely investigated for use in various practical applications such as smart prosthetics, [4] wearable devices, [5] and artificial robot arms [6,7]. For robotic applications, the two fundamental research areas on tactile sensor development connected to object-controlled lifting and grasp functioning with the ability to describe diverse surface textures have developed continuously [8].

In the last decade, various types of pressure sensors, including resistive, [9] capacitive, [10] triboelectric [11] and piezoelectric [12] sensors, have been intensively examined as promising candidates for tactile sensing applications. Among all sensor types, piezoelectric-based pressure sensors are autonomously powered, which generate a consistent electrical signal in response to applied force/pressure. Hence the piezoelectric-type sensor represents a prominent, stand-alone and self-reliant sensor, which makes it suitable for robotic tactile sensing to provide reliable electrical inputs for robotic manipulators [13].

With the development of materials science and versatile fabrication techniques, two-dimensional (2D) and three-dimensional (3D) resilient nanostructures have been synthesized sophisticatedly and are receiving greater attention among researchers [14]. In this direction, electrospinning fabrication methods as shown in Figure 1c, have received significant consideration to produce polymer-based one-dimensional functional nanofibers, as shown in Figure 1d [15,16].

Micrometer/sub micrometer electromechanical polymer fibers are promising components for tactile sensing application due their high mechanical flexibility, low weight, and excellent breathability [17]. These polymer-based sensors primarily comprise piezoelectric polymers, including poly(vinylidene fluoride) (PVDF) [18], poly(vinylidenefluoride-co-trifluoroethylene) (PVDF-TrFE) [19] and Barium titanite [20]. Although piezoelectric polymer fibers have already been demonstrated as pressure sensors by various research groups, piezoelectric polymers are expensive. Hence, there are challenges for low-cost manufacturing and large-area fabrication which limits the practical application. On the other hand, converse electromechanical responses were observed inadvertently with electrospun submicron/micron fiber mats composed of nonpiezoelectric polymers, such as poly(D,L-lactic acid (PDLLA), [21] poly(methyl methacrylate) (PMMA) [21], and their composites [22]. These non-piezoelectric materials exhibit piezoelectric properties like conventional piezoelectric materials, indicating a significant high apparent piezoelectric d constant $\left(d_{app}\right)$, as high as 8500 pm V^−1^ for an individual material [21] and 29,000 pm V^−1^ for a composite material [22]. These papers also reported that these excellent electromechanical properties are partly or mainly attributed to the unique electrically charged and mechanically soft nature of the electrospun fiber mats. Observing piezoelectric properties in non-piezoelectric material is a fairly new concept, for instance, D. Hassan, et al. demonstrated piezoelectricity using polystyrene-copper oxide (PS-CuO) nano composites. The study suggested that the (PS-CuO) nanocomposites are highly sensitive for pressure and the electrical resistance of nanocomposites decreases with the increase in pressure [23]. Similarly, Y. Ishii, et al. demonstrated the piezoelectric properties in polystyrene polymers (Figure 1b) and explored the origin of the piezoelectric response. The apparent piezoelectric constant of the fiber mat was measured as 950–1400 pC/N with an applied load of 0.05–0.28 N using the quasistatic method [24]. Conventionally constructing piezoelectric sensors using polymers/composites or crystals seems to be expensive due to base material and fabrication cost. Meanwhile, polystyrene holds a promising future due to its inexpensive base material, low cost fabrication method and large area fabrication. Figure 1e–g represents the device-operating mechanism; Figure 1e denotes that the sensor has zero output potential at an ideal state. Figure 1f shows that the sensor exhibits a positive potential at the terminals when a force is applied. Similarly, Figure 1g represents a negative potential at the electrode terminal when force is removed from the sensor. In this work, we have fabricated atactic polystyrene (aPS) microfibers using an electrospinning method. An as-spun aPS fiber mat was characterized using a field-emission scanning electron microscope. Subsequently, we constructed a pressure sensor using aPS microfibers; electrical and mechanical characteristics were studied. Finally, the pressure sensor was attached to a commercially available Kuka robotic arm structure for tactile sensing application.

## 2. Materials and Methods

### 2.1. Materials and Fabrication of Fibers

Atactic polystyrene (aPS) was chosen as the base material of electrospun fibers because it is a widely used and inexpensive nonpiezoelectric polymer. aPS (*M*_w _ ≈ 280,000, Sigma-Aldrich, Saint Louis, Missouri, USA,) was dissolved in *N*,*N*-dimethylformamide (DMF) with a concentration of 30 wt% at room temperature. Next, the solution was electrospun with commercial apparatus (NANON, Mecc, Ogari-shi, Fukuoka, Japan). Details of the electrospinning conditions are as follows: the solution was fed with a constant rate of 5.0 mL h^−1^ and discharged through a single stainless steel needle (inner diameter of 0.59 mm); here, the needle moved reciprocally in a straight line with a width and speed of 60 mm and 10 mm s^−1^, respectively, to produce the electrospun fibers in a wide area. A metallic drum with a diameter and width of 210 mm and 660 mm, respectively, was placed 17 cm below the tip of the needle to support the following collector sheet on which the electrospun aPS fibers were deposited. The collector sheet was composed of a thin polypropylene (PP) sheet (25 mm × 310 mm with a thickness of 0.15 mm), on one side of which a conductive carbon tape (20 mm × 310 mm with a thickness of 0.11 mm) was attached to enhance the adhesion of the electrospun fibers as shown Figure 2a. The collector sheet was attached to the surface of the drum; then, the carbon tape and drum had electrically connected to each other and were grounded; 12.0 kV was applied to the needle and the electrospun aPS fibers were deposited for 20 min. The rotating speed of the drum was 50 rpm.

### 2.2. Sensor Structure

The schematic and pictures of the sensor developed in this study is shown in Figure 2b–d. Its operation mechanism as a pressure sensor is basically explained using the electret condenser model reported recently [17]. Briefly, the as-electrospun aPS fiber mat was deposited on the bottom electrode, which is the conductive carbon tape in the present study, uniquely holds space charges with both positive and negative polarities [24]. In addition, the positive and negative space charges are generally distributed in the upper and lower parts of the fiber mat, respectively. These unique space charges induce compensation charges in the bottom and upper electrodes. When the distance between the bottom and upper electrodes change due to application of an external pressure to the sensor and a resulting deformation of the sensor, the amount of the space charges in a steady state change. This change in the charge amount in the steady state outputs charges so that pressing and releasing the sensor generate charges and output voltage.

The collector sheet, on which the aPS fiber mat was deposited, cut to the size of 25 mm × 30 mm and attached to the PET plate (25 mm × 30 mm with a thickness of 0.7 mm) and used as the bottom part as shown in Figure 2b. An aluminum foil (20 mm × 30 mm with a thickness of 0.1 mm) was used as a upper electrode, which was attached on a PP sheet (25 mm × 30 mm with a thickness of 0.15 mm) and PET plate (25 mm × 30 mm with a thickness of 0.7 mm). The multi-layered PP sheet in Figure 2b was composed of thin PP sheets with the width and thickness of 25 mm and 0.15 mm, respectively. The multi-layered sheet kept the upper and bottom electrodes uncontacted and deformed when the external pressure was applied to the sensor. The PET stopper (20 mm/15 mm × 1 mm with a thickness of 0.7 mm) was attached to the upper plate to prevent the upper electrode from hardly touching and damaging the aPS fiber mat on the bottom electrode. The conducting carbon tape on the multi-layered PP sheet and medical paper shown in Figure 2b were introduced to reduce the generation of contacting charges when the sensor contacted to an object and released it.

### 2.3. Characterization

The shape of the aPS fiber mat was characterized using a field-emission scanning electron microscope (FESEM; S-4300, Hitachi, Marunouchi, Tokyo, Japan) after a 6-nm-thick gold coating. Figure 3a shows the top view of the FESEM image of the as-electrospun aPS fiber mat. The fibers have smooth morphologies without beaded morphologies and orients to random directions. The average diameter is determined to be 5.89 ± 0.39 µm (all errors in this paper represent the standard deviation) from 100 points of diameters measured from the FESEM images. Figure 3b shows the cross-sectional FESEM image of the fiber mat. The aPS fibers stack with air spaces and form a fiber mat. The thickness of the fiber mat is approximately 210 µm. Quantitative testing of the electrical outputs of the sensors was performed using the handmade setup shown in Figure 4. Here, the sensor was attached to the L-shaped metallic holder and then it was lowered with a constant speed of 4.0 mm s^−1^ through an X-Axis dovetail feed screw (XLSL120-2, Misumi, Bunkyo-ku, Tokyo, Japan) and DC motor (HG37-30-AB-00, Nidec Copal Electronics, Tokyo, Japan). The applied load to the sensor was measured with the gravimeter (ACS5000, Kyoto, Japan); the load was controlled at the same time as lowering the sensor. The applied load was converted to pressure, dividing the load by the bottom area of the sensor, 6.25 × 10^−4^ m^2^. The initial height of the bottom of the sensor was fixed 5 mm from the surface of the gravimeter as shown in Figure 4a. The time-dependent open circuit voltage, *V*_oc_(*t*), and short circuit current, *I*_sc_(*t*), were measured with a source meter (Model 2450, Keithley, Beaverton, Oregon, USA). Here, *t* represents time.

## 3. Results

### 3.1. Electrical Characteristics

Figure 5a shows *V*_oc_(*t*) from the sensor when the sensor was loaded with different values of pressures. *V*_oc_(*t*) was also measured when the pressure was released, as shown in Figure 5b. The open circuit voltage is generated when the sensor is both loaded and released, which demonstrates that the present sensor acts as a pressure sensor. The absolute values of *V*_oc_(*t*) in each peak increases with increasing the applied pressure, as also plotted in Figure 5c. This result shows that the applied pressure can be identified from the open circuit voltage of the sensor on the condition that the speed for applying pressure is a constant value. The maximum open circuit voltage reaches approximately 3.1 V with the pressure of 19.9 kPa. Although the present study fixed the speed for applying pressure at 4.0 mm s^−1^, the open circuit voltage can increase with an increase in the speed. The maximum *V*_oc_(*t*) with the applied pressure ≥19.9 kPa is saturated because the PET stopper touches the bottom substrate so that the distance between the bottom and the upper electrodes cannot change with the pressure ≥19.9 kPa.

Figure 5d,e show *I*_sc_(*t*) from the sensor when the sensor was loaded and released with/from different values of pressures, respectively. The short circuit current is also generated by loading and releasing the pressure. In addition, the absolute values of *I*_sc_(*t*) in each peak increases with an increase in the applied pressure, as also plotted in Figure 5f, which is similar to the result of the open circuit voltage. The *I*_sc_(*t*) when the sensor was loaded was integrated with the time of loading (Δ*T*) using the following equation:(1)$$\int_{}^{\Delta T}I_{\mathrm{sc}}\left(t\right)\;dt=\Delta Q$$

Here, Δ*Q* represents the integrated output charge when the sensor was loaded. Δ*Q* changes depending on the distance between the bottom and upper electrodes as reported recently [17], such that Δ*Q* changes depending on the value of the applied pressure in the present sensor. Here, the *V*_oc_(*t*) and *I*_sc_(*t*) from the present sensor changes depending on the speed for applying pressure; however, Δ*Q* can be constant even when the speed changes. Figure 6a shows Δ*Q* with different applied pressures. Δ*Q* increased with the increasing applied pressure. Hence, the applied pressure can be identified from the Δ*Q* even when the speed for applying the pressure changes.

The apparent piezoelectric *d* constant (*d*_app_) was calculated from the following equation:(2)$$d_{\mathrm{app}}=\Delta Q/F$$
where, *F* is the load applied to the sensor. Figure 6b shows *d*_app_ with different applied pressures. The *d*_app_ is roughly constant for the applied pressure with the range from 11.1 pC N^−1^ to 15.7 pC N^−1^; the average *d*_app_ is 12.9 ± 1.8 pC N^−1^. A detailed comparison is presented in the Table 1, representing electromechanical characteristics and fabrication parameters for non-piezoelectric polymers.

### 3.2. Mechanical Characteristics

The height of the sensor was measured when different pressures were applied to the sensor (Figure 7a). The initial height of the sensor is approximately 10.5 mm; the height is almost maintained up to the applied pressure of 3.4 kPa, which should be considered as the threshold pressure until the sensor starts to deform.

With the pressure range from 4.0 kPa to 17.9 kPa, the height decreases with the increasing applied pressure, which represents the sensor deforms increasing with the pressure. With a pressure of ≥19.9 kPa, the height shows an almost constant value, which is because the PET stopper touches the bottom substrate, so that the distance between the bottom and upper electrodes cannot change.

Figure 7b shows the stress–strain plots for the present sensor. Here, the strain was calculated by dividing the decrement in the height of the sensor by the initial height of the sensor. As observed in Figure 7a, the threshold stress with the stress of ≤3.4 kPa and the upper limit of the strain with the strain of around 0.27 was observed. On the other hand, at the strain range from 0.02 to 0.25, plots are modestly fitted with a linear function. From the tilting of the fitted linear function, the elastic modulus of the sensor was evaluated to be approximately 47.7 kPa. In this study, the multi-layered PP sheet was used to keep the upper and bottom electrodes uncontacted; however, the sheet can be replaced by other materials and structures so that the elastic modulus of the sensor can be simply controlled. The controllable elastic modulus will expand the application fields. In addition, a lower elastic modulus of the sensor with softer structures provides a higher *d*_app_.

## 4. Application

Commercially available tactile sensors such as thin film, capacitive, optical sensors, and magnetic sensors are expensive due to complex fabrication technologies. Moreover, the practical application is inhibited due to the demanding computational power to perform the manipulation process. Meanwhile, the electrospinning fabrication method represents a huge advantage when compared to other fabrication methodologies due to the ease of fabrication and mass production at room temperature. Furthermore, the discovery of piezoelectricity in polystyrene could pave new paths to new research directions and commercial usages. The fabricated pressure sensor (25 × 25 × 7 mm) as shown in Figure 2c,d was connected to a commercially available Kuka LBR IIWA14 R820 model to mimic the tactile sensing application [25]. The Kuka robotic arm was programmed to grab and release an aluminum bar weighing 400 g. The Kuka robotic arm has various programmable pinching forces (force applied to the object to lift); however, we set it to 40 N to lift the aluminum bar. The pressure sensor attached to the arm as shown in Figure 8a–c represents the aluminum bar being grabbed and released. Corresponding open circuit voltages were measured using a DC source meter, and a graph was plotted as depicted in Figure 9. The sensor generated +13 V and −13 V output voltages during a grab and release operation, and the output voltage response indicates the steady stage operation in the experiment. From graph 9, we can conclude that the pressure sensor has a young modulus of 47.7 kPa and will generate a maximum of 13 V. Conversely, this result may be varied if the device’s young modulus has altered or when a greater pinching force is applied. In summary, this study suggests that the pressure sensor built using a low-cost polystyrene polymer material by the electrospinning method can be used as tactile sensors for robots and could replace commercially available expensive pressure sensors.

## 5. Conclusions

Tactile sensors are vital components to emulate the properties of human skin or electronic skin. Regardless, tactile sensors such as capacitive, optical, magnetic, and piezoelectric types are expensive due to high fabrication costs. On the other hand, converse electromechanical responses were observed inadvertently with electrospun submicron/micron fiber mats composed of nonpiezoelectric polymers, such as poly(d,l-lactic acid (PDLLA), [21] poly(methyl methacrylate) (PMMA) and recently polystyrene (PS). In this article, we demonstrated the piezoelectric effect from atactic polystyrene (aPS) material and fabricated microfibers using an electrospinning method. Surface morphology of microfibers were investigated by FSEM. A pressure sensor was fabricated and the output voltage and current responses were calculated using a DC source meter with respect to different applied pressures. A maximum open circuit voltage of 3.1 V was generated at 19.9 kPa applied pressure and the apparent piezoelectric *d* constant (*d*_app_) was calculated as 12.9 ± 1.8 pC N^−1^. The mechanical characteristics of the sensor were evaluated and the stress–strain plot is presented. A fabricated pressure sensor was attached to a commercially available robotic arm to mimic the tactile sensing and the corresponding voltage output (13 V) was estimated to lift the 400 g aluminum bar. In summary, combining an electrospinning fabrication method with a low-cost polystyrene polymer material to fabricate a pressure sensor could pave new paths and new research directions and commercial applications.

## Figures and Tables

**Figure 1 nanomaterials-11-01320-f001:**
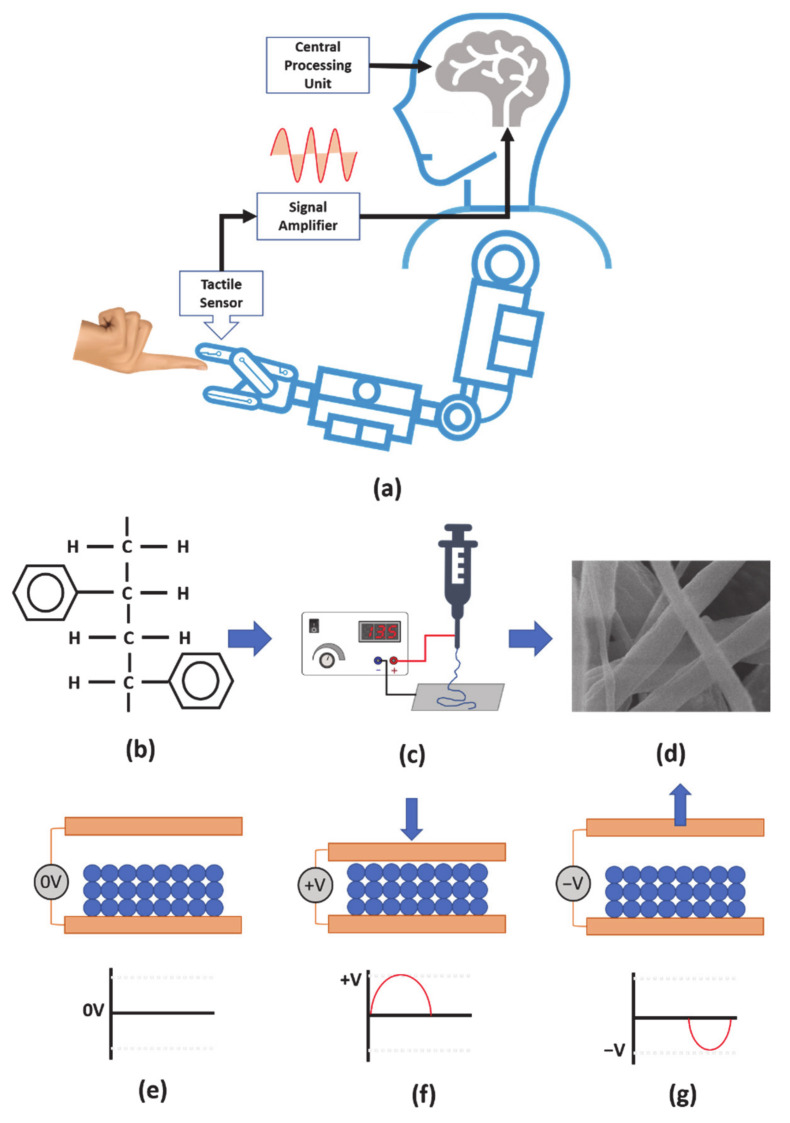
(**a**) Schematic illustration of a robot touch sensation. (**b**) Atactic polystyrene chain. (**c**) Electrospinning set-up. (**d**) SEM image of nanofibers. Schematics of tactile sensor (**e**) neutral, (**f**) compressed and (**g**) released.

**Figure 2 nanomaterials-11-01320-f002:**
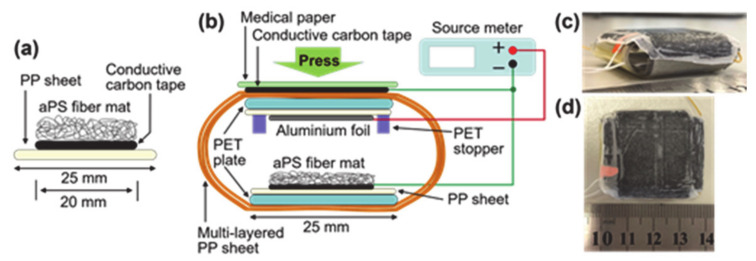
Schematics of (**a**) the collector sheet and (**b**) developed pressure sensor. Photographs of the sensor: (**c**) side-view and (**d**) top view.

**Figure 3 nanomaterials-11-01320-f003:**
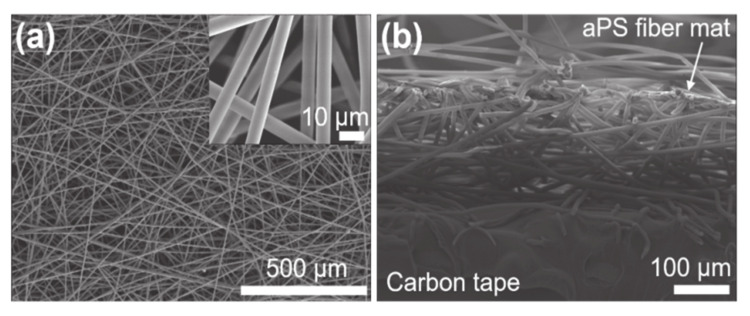
FESEM images of the aPS fiber mat: (**a**) top view and (**b**) cross-section view. Inset in (**a**) shows the enlarged FESEM image of the fiber mat with the top view.

**Figure 4 nanomaterials-11-01320-f004:**
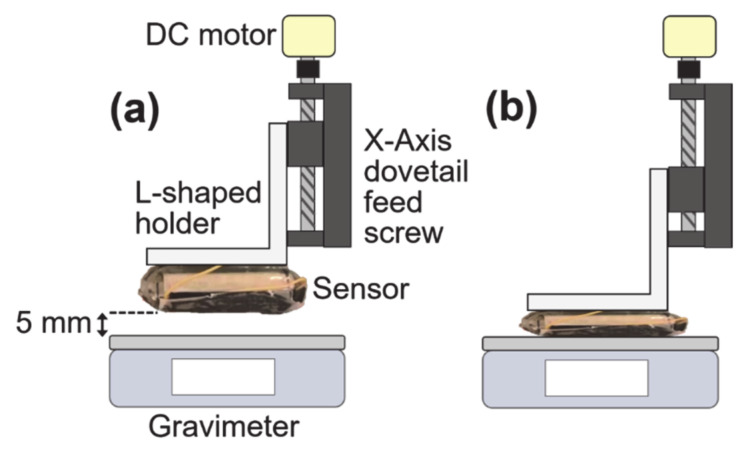
Schematics of the quantitative testing method for the electrical outputs of the sensor: (**a**) the initial state and (**b**) loading state.

**Figure 5 nanomaterials-11-01320-f005:**
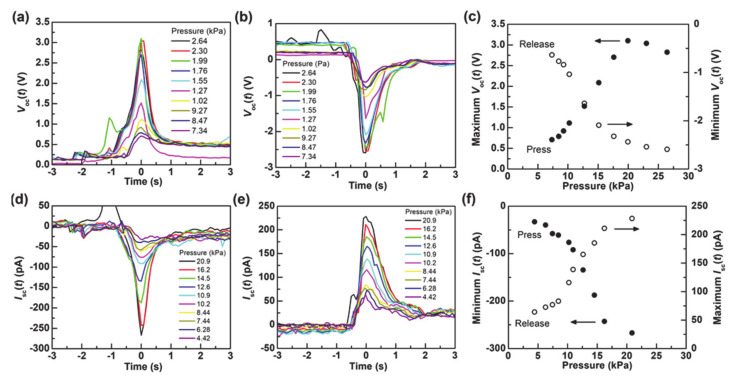
(**a**,**b**) *V*_oc_(*t*) when the sensor was (**a**) loaded and (**b**) released. (**c**) The maximum/minimum *V*_oc_(*t*) with different applied pressures. (**d**,**e**) *I*_sc_(*t*) when the sensor was (**d**) loaded and (**e**) released. (**f**) The maximum/minimum *I*_sc_(*t*) with different applied pressures.

**Figure 6 nanomaterials-11-01320-f006:**
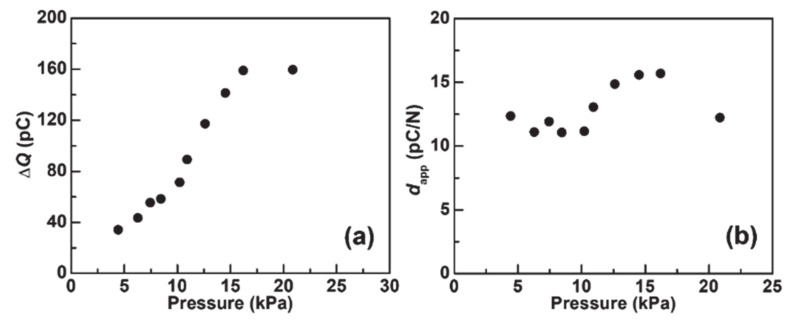
(**a**) Δ*Q* with different applied pressures. (**b**) *d*_app_ with different applied pressures.

**Figure 7 nanomaterials-11-01320-f007:**
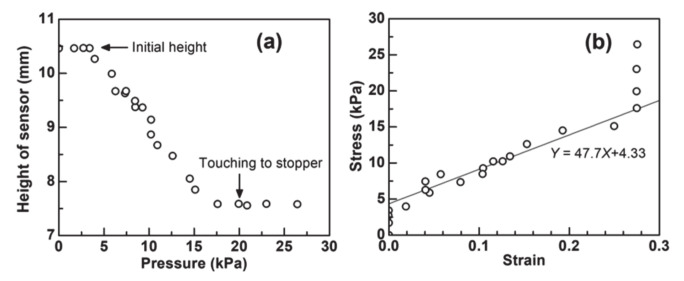
(**a**) Height of the sensor when a different amount of pressure was applied to the sensor. (**b**) Stress–strain plots for the sensor.

**Figure 8 nanomaterials-11-01320-f008:**
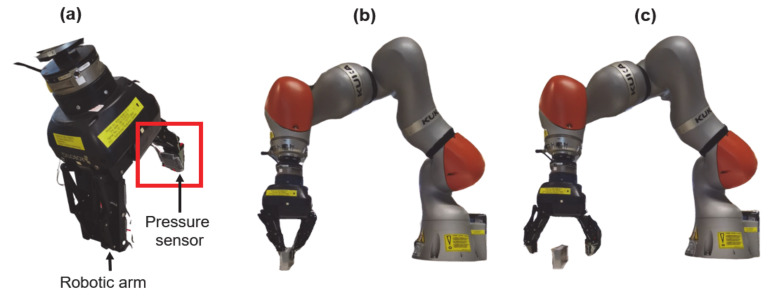
(**a**) Pressure sensor attached to the robotic arm; robotic arm (**b**) grab and(**c**) release mechanism.

**Figure 9 nanomaterials-11-01320-f009:**
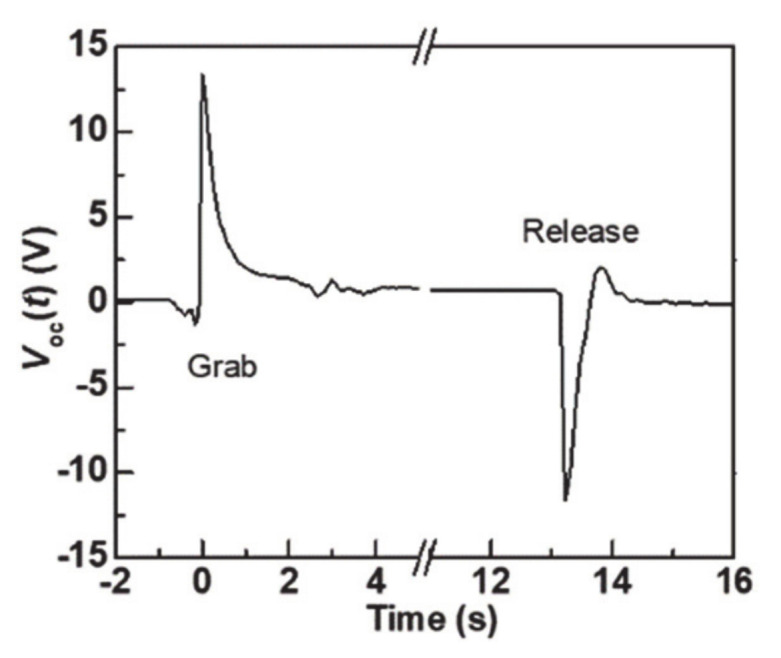
Open circuit voltage with time when the robotic arm grabbed and released the weight.

**Table 1 nanomaterials-11-01320-t001:** Comparison of different non piezoelectric materials fabricated via electrospinning and the electromechanical characteristics.

Materials	Average Fiber Size	Electrospinning Parameters		Principle	Electromechanical Properties	Reference
poly(methyl methacrylate) (PMMA)	1 µm	Rate	0.1 mL/h	Actuation	Piezoelectric constant (d_T_) = 8.5 nm/V	[21]
		Syringe size	0.18 mm			
		Applied Voltage	8 kV			
		Distance	10 cm			
polymer poly(DL-lactic acid) (PDLLA)	0.4 µm	Rate	0.04 mL/h	Actuation	Young modulus = 1.5 kPa Piezoelectric constant = 29,000 × 10^−12^ m V^−1^	[22]
		Syringe size	0.18 mm			
		Applied Voltage	4 kV			
		Distance	10 cm			
atactic polystyrene (aPS)	5.8 µm	Rate	5.0 mL/h	Sensing	Apparent piezoelectric *d* constant (*d*_app_) = 12.9 ± 1.8 pC N^−1^ Young modulus = 47.7 kPa	This study
		Syringe size	0.59 mm			
		Applied Voltage	12 kV			
		Distance	17 cm			

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
