# Peer review of "Fabrication of Pressure Sensor Using Electrospinning Method for Robotic Tactile Sensing Application"

_nanomaterials, 2021, doi:10.3390/nano11051320_

Round 1
Reviewer 1 Report
The manuscript seem to overall be sound and suitable. However, I ask the authors to discuss a clarification and/or deeper explanation on the thermal effects in the nanofibers that may greatly impact the predicted piezoelectric properties in non-piezoelectric material for fabricated pressure sensor. Similarly, to what has already been discussed for different condition in the paper [1]. This needs to be cited where the statement is reported in paragraph Application, from line 236 to 238.
[1]. R. Araneo et al. Thermal-electric model for piezoelectric ZnO nanowires. Nanotechnology, vol.26: 26; 265402. 2015
With these recommendation covered to participate in the improvement of this work, I will agree with the publication of the manuscript
Author Response
Dear Reviewer,
Thank you for your valuable time and inputs. Attached pls find the document with amendments and comments based on your suggestion.

Reviewer 2 Report
Electrospun fiber mat based sensors are finding new areas of applications and the present paper tries to develop sensors for robotic pressure sensing applications. Following are some comments that direct the manuscript to be considered for a major revision:
- The article seems to report the use and application of spun fiber mat at the micro-level. It does not fall under the journal's scope which considers any topic related to the nano level. A comparison of nanofiber-based electrospun mat sensor properties reported in the literature with the microfiber-based sensor presented in this paper should be included in Section 3 to make it relevant.
- Instead of a detailed discussion on materials for pressure sensing in literature, focusing on the comparison of spun fiber mats for pressure sensing would have added value. A table explaining the method, fiber details, electrospinning details, and final electromechanical properties will add value to the manuscript. Also, distinguish nanofibers from microfibers in the table.
- The contribution is low for publication in a journal. The concept of spun fiber mat is not new. Refs [13] and [17] have introduced it already. The challenges associated with the reported fiber mat sensors to be used as robotic tactile applications needs to be explained and emphasized in the introduction. The developed mat sensors have been claimed to have better properties when compared to reported sensors, Therefore, to add a novelty, better properties need to be explained with comparison to other sensors and with valid reasons.
- Next, repeatability and fatigue curves with typical robotic pressure need to be obtained from additional experiments and presented in the contributing sections as new contributions. This is very much needed as the mat has breathable fibers with low structural integrity and may fail under repeated loading in industrial robotic applications.
Author Response
Dear Reviewer,
Thank you very much for your recommendations, inputs, and time for reviewing our article. Attached pls find the article with the latest amendments and suggestions.

Reviewer 3 Report
The manuscript is generally well organized. I have the following comments for the authors:
- What is the uniqueness of the proposed pressure sensor compared with the existing sensors?
- How is it can be used for the practical applications?
Author Response
Dear Reviewer,
Thank you very much for your time. Attached please find our inputs for your valuable comments

Round 2
Reviewer 2 Report
None